# Integration of Whole-Genome Resequencing and Transcriptome Sequencing Reveals Candidate Genes in High Glossiness of Eggshell

**DOI:** 10.3390/ani14081141

**Published:** 2024-04-09

**Authors:** Xiang Song, Shuo Li, Shixiong He, Hongxiang Zheng, Ruijie Li, Long Liu, Tuoyu Geng, Minmeng Zhao, Daoqing Gong

**Affiliations:** 1College of Animal Science and Technology, Yangzhou University, Yangzhou 225009, China; mx120210868@stu.yzu.edu.cn (X.S.); liulong@yzu.edu.cn (L.L.); tygeng@yzu.edu.cn (T.G.);; 2Jiangsu Beinongda Agriculture and Animal Husbandry Technology Co., Ltd., Taizhou 225300, China

**Keywords:** whole-genome resequencing, RNA-seq, eggshell gloss

## Abstract

**Simple Summary:**

Eggshell gloss is an important characteristic for the manifestation of eggshell appearance. However, the reason for differences in eggshell glossiness is still unclear. The aim of this study is to perform a preliminary investigation into the formation mechanism of eggshell gloss and to identify potential genes through high-throughput sequencing. *HTR1F*, *ZNF536*, *NEDD8*, *NGF* and *CALM1* were identified as potential candidate genes that may affect eggshell gloss, which provide a reference for the study of eggshell gloss and lay a foundation for improving egg glossiness in layer breeding.

**Abstract:**

Eggshell gloss is an important characteristic for the manifestation of eggshell appearance. However, no study has yet identified potential candidate genes for eggshell gloss between high-gloss (HG) and low-gloss (LG) chickens. The aim of this study was to perform a preliminary investigation into the formation mechanism of eggshell gloss and to identify potential genes. The eggshell gloss of 300-day-old Rhode Island Red hens was measured from three aspects. Uterine tissues of the selected HG and LG (*n* = 5) hens were collected for RNA-seq. Blood samples were also collected for whole-genome resequencing (WGRS). RNA-seq analysis showed that 150 differentially expressed genes (DEGs) were identified in the uterine tissues of HG and LG hens. These DEGs were mainly enriched in the calcium signaling pathway and the neuroactive ligand–receptor interaction pathway. Importantly, these two pathways were also significantly enriched in the WGRS analysis results. Further joint analysis of WGRS and RNA-seq data revealed that 5-hydroxytryptamine receptor 1F (*HTR1F*), zinc finger protein 536 (*ZNF536*), NEDD8 ubiquitin-like modifier (*NEDD8*), nerve growth factor (*NGF*) and calmodulin 1 (*CALM1*) are potential candidate genes for eggshell gloss. In summary, our research provides a reference for the study of eggshell gloss and lays a foundation for improving egg glossiness in layer breeding.

## 1. Introduction

Eggs are widely used as cheap but nutritious food or an ingredient in food products [1]. The appearance traits of eggs are important factors influencing consumers’ buying inclination. In addition to numerous eggshell appearance qualities (shape, color, etc.), eggshell brightness is also an important characteristic reflecting eggshell appearance [2]. However, most studies on eggshell appearance traits have neglected eggshell gloss [3]. Therefore, it is vital to explore the formation mechanism of eggshell brightness.

The brightness of an egg’s surface can be measured in terms of gloss [4]. Gloss is related to the ability of the eggshell surface to reflect light directly [5]. The glossy appearance of eggshells is produced by an extremely smooth cuticle, and optical calculations have demonstrated that surface smoothness is the major reason for the production of gloss [6]. The gloss in the equator of an eggshell can reflect the gloss of the entire egg, which makes it the best point at which to measure the gloss of an egg [7]. It is now known that eggshell gloss is affected by multiple factors, including the species (genetic differences), age, health condition [8,9] and other environmental factors including nutrition and digestion.

Approximately 24 h are required for an egg to form in the oviduct of a chicken, and the shell is formed in the uterus, which takes about 20 h [10]. The cuticle, which determines the glossiness of the eggshell, is formed about 2–3 h before laying. It has been reported that the expression of genes in the uterus will affect the quality of eggshell. Previous studies have used RNA-seq analysis to identify numerous genes that exhibit high expression levels in the chicken uterus [11,12]. More than 600 genes are differentially expressed in the uterus during eggshell formation [13]. Some of these uterine genes proved to be useful as biological markers for genetic improvements in phenotypic traits [14]. Whole-genome resequencing (WGRS) provides an approach to explore the genomic variations and lays the foundation for further functional analysis. WGRS has been widely used in the field of livestock and poultry breeding. Studies have used this technology to screen candidate genes for chicken fertilization rate and egg production rate [15], to detect variants associated with economic traits [16] and to assess the patterns of different locations of variation and linkage disequilibrium in commercial chicken populations [17].

However, there are few studies on eggshell gloss, and the molecular mechanism of its formation is not clear. Therefore, in the present study, eggshell glossiness was measured comprehensively, based on which, two groups of hens producing eggs with high and low eggshell gloss (HG and LG) were selected for sample collection. We hypothesized that eggshell gloss formation is determined by the smoothness of the eggshell surface (cuticle), and genetic differences involved in cuticle formation can lead to different gloss levels in eggs laid by different hens. Therefore, WGRS and RNA-seq were performed using the collected blood samples (DNA required for WGRS can be easily extracted from blood samples) and uterine tissues, respectively, in order to screen potential candidate genes associated with eggshell glossiness.

## 2. Materials and Methods

### 2.1. Birds and Sample Collection

The experimental animals were 300-day-old Rhode Island Red hens, which were provided by Jiangsu Beinongda Agricultural Animal Husbandry Technology Co., LTD. First, the eggshell glossiness of the whole flock (*n* = 1127) was manually observed for three consecutive days. Then, 40 hens were selected and divided into two groups (*n* = 20) of high gloss (HG) and low gloss (LG) according to their eggshell gloss. Based on manual observations and instrumental measurements, five hens with the highest and lowest eggshell gloss were further selected for sampling from the HG and LG groups, respectively (*n* = 5). Finally, the egg-laying time of these 10 hens was recorded for five days in order to predict the egg-laying time on the sampling day, and uterine tissues and blood samples were collected from these hens 2 h before predicted egg-laying time. Blood samples were stored at −20 °C and uterine tissue samples were stored at −80 °C until use.

### 2.2. Eggshell Quality Measurement

External and internal egg quality traits including eggshell gloss, egg weight (EW), egg shape index, eggshell strength (ES), eggshell thickness (ET) and egg yolk weight (EYW) were measured.

Eggshell gloss was measured from three aspects: (1) Sensory measurement: the glossiness of the eggs was manually observed and scored by two experienced laying hen breeders. The eggs of the whole flock were scored on a scale of 1–5, where 5 represents the best glossiness and 1 the weakest. Based on the grading of the eggs laid by each hen in three days, 20 individuals in which all laid eggs were of grade 5 were selected as candidates for the HG group and 20 individuals with eggs all of grade 1 were selected as candidates for the LG group. (2) Glossmeter measurement: The glossiness values of the eggs laid by the hens in the LG and HG groups were measured with a glossmeter (NOVO-CURVE) at the equator of the eggshell. (3) Microstructure measurement: A scanning electron microscope (SEM) was used to observe the cuticle texture of the eggshell. Pieces of eggshell (1 cm × 1 cm) were cut from the equator of the eggshells of eggs from the HG and LG group. They were mounted on an aluminum stub and gold sputter coated for about 15 min. Thereafter, the eggshells were scanned and photographed under the SEM.

The eggs’ length and width were measured in millimeters using a vernier caliper for eggshell index. Eggshell strength was measured using an FGX-5R eggshell strength tester. The ET was measured (mm) using ETG-1061A on the blunt region, equatorial region and sharp region, and the average was considered the value for the egg. The yolk and albumen were then separated and weighed (g).

### 2.3. Genomic DNA Extraction, RNA Extraction and qPCR Assay

Genomic DNA from 10 selected hens (HG and LG group, *n* = 5) was used for WGRS. The genomic DNA was extracted from blood samples using the FastPure Cell/Tissue DNA Isolation Mini Kit-BOX1 (Vazyme, Nanjing, China) according to the manufacturer’s protocol except for the volume of the samples. As chicken red blood cells contain nuclei, we only used 20 μL blood samples for DNA extraction, rather than the 250 μL recommended by the protocol (the rest of the volume was replaced with PBS). The total RNA of uterine tissues from the HG and LG group (*n* = 5) was extracted using the RNA Easy Fast Tissue/Cell Kit (Vazyme) according to the manufacturer’s protocol.

cDNA was synthesized using HiScript III All-in-one RT SuperMix Perfect for qPCR (Vazyme). Primers were designed using the NCBI Primer Design Tool. cDNA samples were subjected to ChamQ Universal SYBR qPCR Master Mix (Vazyme) according to the manufacturer’s protocol. The 2^−ΔΔCt^ method and internal normalization were used to analyze the quantification results. The information regarding primers used for qPCR amplification is listed in Appendix A.

### 2.4. Transcriptome Sequencing (RNA-seq)

RNA purification, library construction and paired end (PE) sequencing were performed based on an Illumina sequencing platform. The RNA-seq data reported in this study were archived in the NCBI database with the accession number PRJNA981231 (https://www.ncbi.nlm.nih.gov/bioproject/PRJNA981231) (accessed on 7 June 2023). The filtered reads were compared to the reference genome (bGalGal1.mat.broiler.GRCg7b) using Hisat2 (2.2.1) [18] software. HTSeq (2.0.5) [19] was used to calculate the original expression of the gene. Fragments Per Kilobase of transcript sequence per Millions base pairs sequenced (FPKM) was used to standardize the expression levels, and DESeq [20] was used to analyze the differences in gene expression. The conditions for screening differentially expressed genes were as follows: |log2FoldChange| > 1 and *p*-value < 0.05. The pheatmap (1.0.12) R package [21] was used to conduct bidirectional cluster analysis of the union of differential genes and samples of all comparison groups.

### 2.5. Whole-Genome Resequencing

The sequencing library was prepared using the standard library building process of Illumina’s Tru Seq DNA PCR-free prep kit reagent, and the Nova Seq sequencer for 2 × 150 bp double-ended sequencing was adopted at Shanghai Personal Biotechnology Co., Ltd. (Shanghai, China). The WGRS data reported in this study were archived in the GSA database with the accession number PRJNA992581 (https://www.ncbi.nlm.nih.gov/sra/PRJNA992581) (accessed on 14 July 2023). Fastp (v0.20.0) [22] was used for data quality control, and BWA(0.7.12-r1039) [23] was used to compare the filtered high-quality data to the reference genome. GATK (4.4.0.0) [24] software was used to detect genetic variation (GV). Then, the data were further filtered using the following criteria: (1) Fisher test of strand bias (FS) ≤ 60; (2) Haplotype Score ≤ 13.0; (3) Mapping Quality (MQ) ≥ 40; (4) Quality Depth (QD) ≥ 2; (5) Read Pos Rank Sum ≥ −8.0; (6) MQ Rank Sum > −12.5; (7) alternative allele called in ≥ 4 reads. GV were annotated using ANNOVAR (version 24 October 2019) [25] software, and population-specific SNPs were attained. For the SNP analysis, we counted the frequency of the four genotypes (0/0, 0/1, 1/1, /) in the HG and LG groups. Group-specific SNPs were identified in this project by setting a threshold for the frequency of population-specific genotypes, and the HG population-specific SNPs were selected if the frequency of a genotype at these SNPs = 1 in the HG population and the frequency of the same genotype was =0 in the LG population.

### 2.6. Enrichment Analysis

The GO enrichment analysis of genes was conducted by the top GO (3.14) R package. KEGG pathway enrichment analysis was conducted by the clusterProfiler (3.8.1) R package. GO terms and KEGG pathways with *p*-values < 0.05 were considered significantly enriched among genes.

### 2.7. Data Analysis

Population glossiness data were organized using Excel, and statistics analysis was performed using R software (4.1.0). The data obtained were submitted to analysis of variance with the F test. Correlation coefficients between eggshell gloss and other traits were generated using Spearman’s rank correlation. The correlation matrix plot was analyzed with the R package.

## 3. Results

### 3.1. Measurement and Analysis of Eggshell Gloss

Eggs laid by the experimental flocks were graded (grade 1–5) by sensory measurements, and the eggs with the lowest (class 1) and highest (class 5) glossiness were shown in Figure 1A. Based on our records, 915 hens produced eggs with different degrees of glossiness, and 212 hens produced eggs with the same degree of glossiness. According to the results of the sensory measurements, 20 hens were selected as candidate hens for the LG and HG groups from hens laying grade 1 and 5 eggs, respectively. Eggs laid by each hen in the LG and HG groups were measured by a glossmeter, and the results showed that the gloss values of HG eggs were significantly higher than those of LG eggs (Figure 1B). Thereafter, 10 hens which laid eggs with significantly different gloss values (*p* < 0.01) (Figure 1C, Appendix A) were selected from the LG and HG groups (*n* = 5) as candidates for subsequent RNA-seq and WGRS analysis. In addition, the surface texture of HG and LG eggshells was imaged using SEM. The SEM images showed significant differences in the cuticle texture of HG (Figure 1D,F) and LG (Figure 1E,G) eggshells. The surfaces of HG eggshells were smoother than those of LG eggshells.

### 3.2. Population Glossiness Distribution and Correlation Analysis

The distribution of eggshell gloss in the population was statistically analyzed (*n* = 1127), and correlations between glossiness and other egg quality traits were calculated. As shown in Figure 2, the majority of eggshell gloss values were at the intermediate level, showing a trend similar to a binary distribution. In addition, we found that eggshell gloss was not correlated with any other eggshell trait such as egg weight or eggshell strength (Figure 2). These results indicated that eggshell gloss was a relatively stable and independent trait.

### 3.3. RNA-seq Revealed Significant Differences in Gene Expression Patterns Related to Eggshell Gloss

RNA-seq analysis was performed on uterine tissues at the predicted timepoint (2 h before egg laying) to reveal gene expression patterns in HG and LG chickens. Information regarding the quality of the RNA-seq data is listed in Appendix A. Figure 3A shows 99 upregulated DEGs (differentially expressed genes) and 51 downregulated DEGs. Table 1 lists information regarding the 10 DEGs with the lowest *p* values. To determine the reliability of the RNA-seq results, we randomly selected 10 DEGs for qPCR analysis. We found that the qPCR results were consistent with the RNA-seq results, which indicated the reliability of the RNA-seq results (Figure 3B). To further understand the biochemical functions of the DEGs, the 150 DEGs (99 upregulated and 51 downregulated) were used to perform GO and KEGG enrichment analyses. GO terms were classified into biological process (BP), cellular component (CC) and molecular function (MF). In total, 479 GO terms were significantly enriched, and the top 15 terms are shown in Figure 3C (*p* < 0.05). The DEGs were significantly enriched in six KEGG pathways (*p* < 0.05) including the calcium signaling pathway and neuroactive ligand–receptor interactions (Figure 3D).

Weighted correlation network analysis (WGCNA) [26] was used to identify differentially co-expressed modules related to the glossiness. We identified 23 gene modules (Appendix A), of which MEblack was significantly correlated with eggshell gloss (correlation coefficient = 0.874; *p* = 0.01). Hub genes (Appendix A) such as TAGLN were screened in this highly relevant module.

### 3.4. WGRS Analysis to Identify Group-Specific Genetic Variation and Associated Genes

Genomic DNA extracted from the blood samples of five hens each from the LG and HG groups (Figure 1C) was analyzed by WGRS. Appendix A present information regarding quality analysis, mapping rates and average sequence coverage. A large number of genetic variants (151,340 SNPs, 123,110 INDELs, 121,077 CNVs and 141,016 SVs in total) were detected in both groups of samples compared to the reference genome (bGalGal1.mat.broiler. GRCg7b) (see Appendix A for statistical information). Here, we focused on SNP loci and distinguished 15,040 population-specific SNPs (Table 2), which were localized on 1601 genes. Nonsynonymous mutations occurring in exons may affect the structure of the protein and thus lead to altered function, and mutations occurring in UTR5 may have an effect on the expression of genes. Therefore, we focused on these two parts of the SNP (detailed in Appendix A). In addition, GO enrichment analyses were performed on the biological function of these group-specific SNPs located in genes that were enriched in 1053 terms (Figure 4A). KEGG pathway analysis revealed the significant enrichment of five pathways such as the calcium signaling pathway and neuroactive ligand–receptor interaction (Figure 4B). Then, 63 group-specific SNPs were identified based on the frequency of genotype distribution in the population (Appendix A).

### 3.5. Joint Analysis of WGRS and RNA-seq

Based on the above results, we jointly analyzed the WGRS and RNA-seq data to further screen for candidate genes that might affect the glossiness of Rhode Island Red chicken eggs (Figure 5). On the one hand, we compared the genes associated with group-specific SNPs in WGRS with the DEGs screened by RNA-seq and obtained 11 overlapping genes as the first portion of the candidate genes (Table 3). We noticed that zinc finger protein 536 (*ZNF536*) had the lowest *p*-value and the largest number of SNPs. On the other hand, we compared the KEGG pathways that were significantly enriched in the WGRS and RNA-seq analyses and screened two overlapping pathways, the calcium signaling pathway and neuroactive ligand–receptor interactions. Genes in the two pathways were selected as the second portion of the candidate genes (Table 4). Meanwhile, we identified differentially co-expressed modules related to gloss and integrated WGS data to determine if certain SNPs are associated with hub genes in the sub-networks. Protein–Protein Interaction Networks (PPIs) indicated SNPs located in CALM1 are associated with TAGLN. According to the above results, 5-hydroxytryptamine receptor 1F (*HTR1F*) is a differentially expressed gene involved in the neuroactive ligand–receptor interactions pathway. Nerve growth factor (*NGF*) was involved in one of the key pathways and it had multiple SNPs in the UTR5 region. NEDD8 ubiquitin-like modifier (*NEDD8*) had multiple SNPs in exons which are non-synonymous substitutions. Calmodulin1 (*CALM1*) was reported to be involved in eggshell gloss formation [27]. Overall, taking into account the significance level of DEGs, the number of SNPs, the results of KEGG pathways analysis and the literature reviews, *ZNF536*, *HTR1F*, *NGF*, *NEDD8* and *CALM1* were selected as potential candidate genes involved in the eggshell gloss of HG and LG hens. SNP information regarding these genes is listed in Table 5.

## 4. Discussion

Eggshell gloss has an important effect on the eggshell appearance. For wild birds, brightly colored eggs may be a “releaser” signal that attracts males to incubate the eggs [28]. As the gloss and color of tinamou eggs fade during incubation, females can rationally choose laying nests by the glossiness of the eggshells [29]. In recent years, more and more layer breeders have started to choose eggshell gloss as a breeding indicator. However, there is no dedicated equipment and standardized method for measuring eggshell gloss, and the molecular mechanism of its formation and the key genes are not clear, which limit the progress of selection for this trait in layer breeding. Therefore, in this study, we used globally widely bred Rhode Island Red hens as the research object. Firstly, we used different measurement methods to accurately evaluate the eggshell gloss of eggs, based on which, we screened the two-tailed samples (HG and LG) and then analyzed the differences at the genome level and the transcriptome level and finally screened for candidate genes that might be related to eggshell gloss.

For RNA-seq, it is crucial to select the most suitable samples due to the temporal and spatial specificity of gene expression. Indeed, egg laying in poultry is a precisely regulated physiological process. Ovulatory traits are determined by ovarian function and regulated by the hypothalamic pituitary gonadal axis (HPG), and the timing of ovulation, the formation of egg and the final laying time are all highly regular [30]. Calcification of the eggshell occurs in the uterus and results in the formation of a complete eggshell structure through three stages: the initiation of calcification, linear deposition and late calcification [31,32,33]. Since the eggshell gloss trait of interest is closely related to the surface of the eggshell, we hypothesized that “late calcification” is the key stage that affects eggshell gloss. Therefore, we observed the laying time of the candidate hens three days in advance and collected mucosal tissues from the uterus two hours before the predicted laying time for RNA extraction and RNA-seq analysis.

RNA-seq analysis showed that 150 DEGs were identified in the uterus of HG chickens compared to LG chickens. GO and KEGG enrichment analyses of DEGs significantly enriched 479 important GO terms and six KEGG pathways, which were mainly associated with cellular processes such as environmental information processing, the calcium signaling pathway, and neuroactive ligand–receptor interactions (*p* < 0.05). A previous fish study found that the neuroactive ligand–receptor interaction pathway can influence steroid hormone synthesis in the gonads via the HPG axis [34]. The neuroactive ligand–receptor interaction pathway may affect egg production in chickadees through a mechanism similar to that in fish [35,36,37]. Therefore, DEG mapping to the neuroactive ligand–receptor interaction pathway and calcium signaling pathway may play an important role in eggshell gloss formation. Considering shell glossiness is a complex trait influenced by multiple genes, differential co-expression transcriptomic network analysis was performed. We applied WGCNA to analyze the clusters of co-expression genes to screen genes that may be related to eggshell gloss information. The results (Appendix A) may provide a reference for further screening and the validation of candidate genes.

In addition, WGRS analyses revealed substantial inter-group genetic differences between LG and HG. Among them, we focused on and analyzed the identified SNPs. Then, group-specific SNPs were identified based on the genotype distribution frequencies of HG and LG chickens, and the genes where the SNPs were located were analyzed by GO enrichment analysis and KEGG pathway analysis. The results of enrichment analysis generated 1053 significant GO terms and five KEGG pathways. Notably, the calcium signaling pathway and the neuroactive ligand–receptor interaction (*p* < 0.05) pathway were significantly enriched in both WGRS and RNA-seq. This suggests, on the one hand, that our choice of RNA-seq samples was appropriate and, on the other hand, that genomic variation as well as expression differences in these two pathways in the LG and HG groups may be an important mechanism contributing to the differences in eggshell gloss between the two groups.

Finally, we jointly analyzed the WGRS and RNA-seq results to screen candidate genes that might affect eggshell gloss. Calcium is essential for egg formation in laying hens, and it has a significant effect on laying performance [38,39,40]. Calcium is involved in the regulation of androstenedione production and uterine contraction in laying hens [41]. The binding of calmodulin and calcium ions promotes the secretion of steroid hormones from the cervical cells of laying hens [42]. *CALM1* is a prototypical calcium sensor, which is an important gene for reproduction [43]. Bioinformatics analysis revealed that the SNP(g.44069941G  >  A) in *CALM1* affects egg production in chickens [44]. In addition, *CALM1* has been reported to be an important regulator of androgen production in chicken follicular membrane cells [45], which increases the strength of the eggshell to some extent [46]. *HTR1F* is expressed in different regions of the brain and pituitary gland [47]. It may be involved in regulating the release of prolactin from the chicken pituitary and influences egg production by modulating ovarian metabolic function [48]. Previous studies have found that *ZNF536* may affect eggshell weight in chickens [49]. *NEDD8* is a ubiquitin-like protein that controls important biological events by linking to members of the cullin family [50]. *NEDD8* was found to affect the proliferation and apoptosis of bovine follicular granulosa cells [51]. *NGF*, the first member of the neurotrophin family to be isolated from nervous tissue, is a major mediator in the regulation of nerve growth, proliferation, differentiation and survival [52]. In addition, *NGF* plays an important role in neurodegenerative diseases and neuronal survival [53]. Therefore, we preliminarily identified a total of five candidate genes, *HTR1F*, *ZNF536*, *NEDD8*, *NGF* and *CALM1*, based on gene expression, SNP number, SNP location, SNP-associated hub genes and a literature review. However, although we preliminarily identified five candidate genes that may contribute to the difference between HG and LG eggshells, some limitations in this study should be taken into consideration. The mechanism of the functions of the candidate genes needs to be further investigated, and whether the SNP loci of these genes can be applied to molecular marker-assisted selection also needs to be validated in a larger population.

## 5. Conclusions

In conclusion, this is the first study using RNA-seq (uterine tissue) and WGRS to screen for candidate genes that may contribute to the gloss differences between HG and LG eggshells. RNA-seq and WGRS analyses revealed significant differences in gene expression patterns between HG and LG groups and a large number of inter-group differential SNPs. GO and KEGG analyses showed that the related genes were predominantly enriched in calcium signaling pathways and neuroactive ligand–receptor interactions. In addition, five candidate genes that may affect the glossiness of HG and LG eggs were screened by combined WGRS and RNA-seq analysis. This study provides a reference for the study of eggshell gloss and lays a foundation for improving egg glossiness in layer breeding.

## Figures and Tables

**Figure 1 animals-14-01141-f001:**
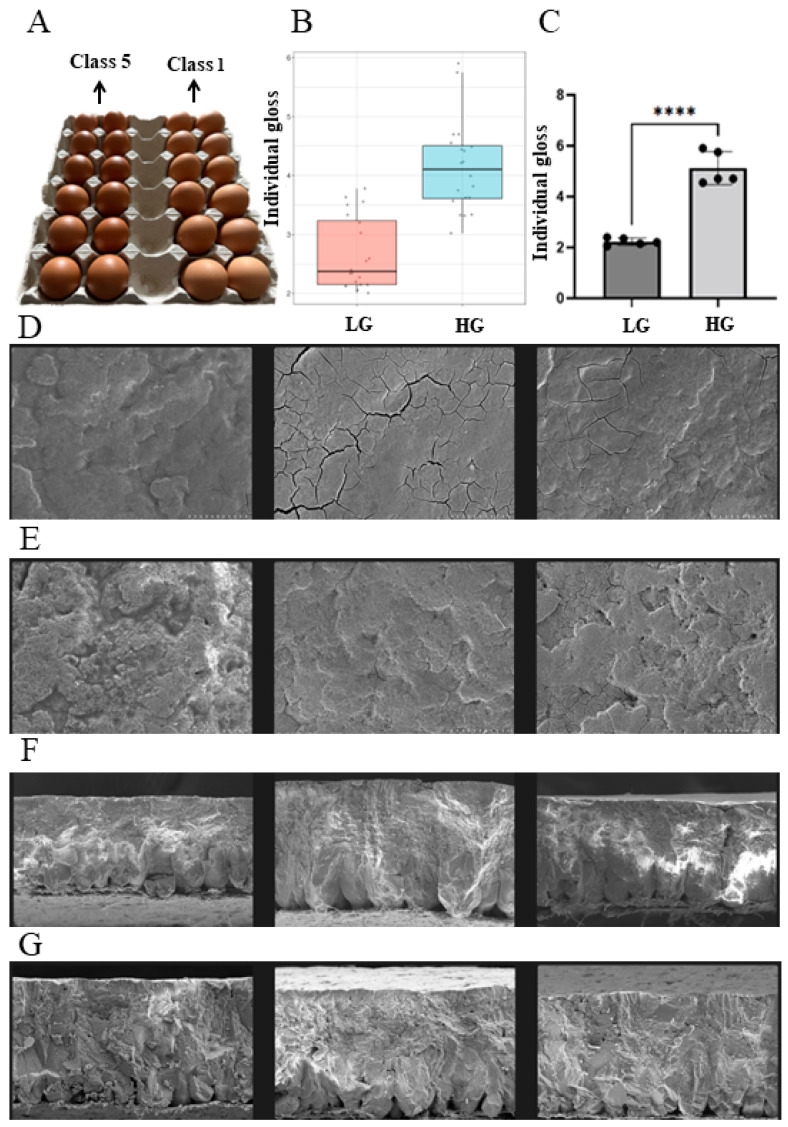
(**A**) HG (high-gloss) eggs (class 5) and LG (low-gloss) eggs (class 1). (**B**) Boxplot of candidate chickens’ gloss value (*n* = 20). (**C**) Individual gloss values of eggs of HG and LG hens used for RNA-seq and WGRS (*n* = 5). Symbol “****” indicated a significant difference at *p* < 0.0001. (**D**–**G**) Scanning electron microscope (SEM) images of HG and LG eggshells. (**D**) HG eggshell surface. (**E**) LG eggshell surface. (**F**) HG eggshell intersecting surface. (**G**) LG eggshell intersecting surface. Scale bars for SEM images: 100 μm (**D**,**E**), 200 μm (**F**,**G**).

**Figure 2 animals-14-01141-f002:**
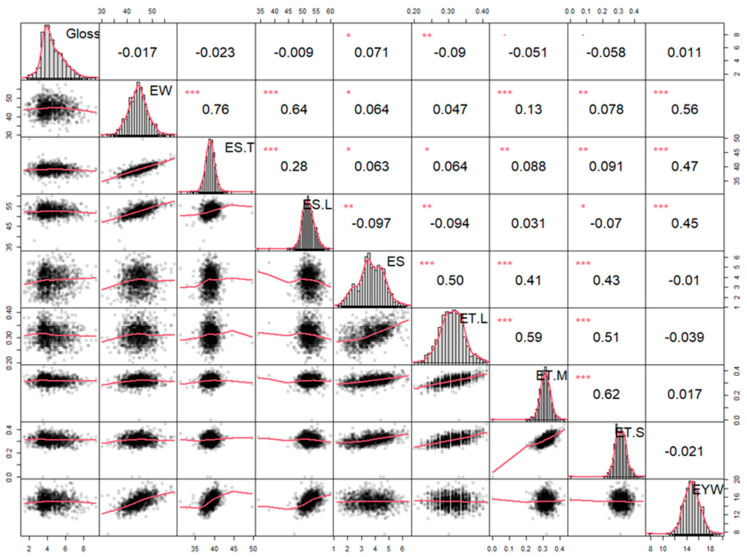
The correlation matrix plot of eggshell gloss among different eggshell traits. On the diagonal are the univariate distributions, plotted as histograms and kernel density plots. On the right of the diagonal are the phenotypic pairwise correlations, with red stars indicating significance levels (*, *p* < 0.05, **, *p* < 0.01, ***, *p* < 0.001). On the left side of the diagonal is the scatter-plot matrix, with LOESS smoothers in red to illustrate the underlying relationship. Abbreviations: EW: egg weight; ES. T: egg shape index—transverse diameter; ES. L: egg shape index—longitudinal diameter; ES: eggshell strength; ET. L: eggshell thickness—blunt end; ET. M: eggshell thickness—equatorial part; ET. S: eggshell thickness—sharp end; EYW: egg yolk weight.

**Figure 3 animals-14-01141-f003:**
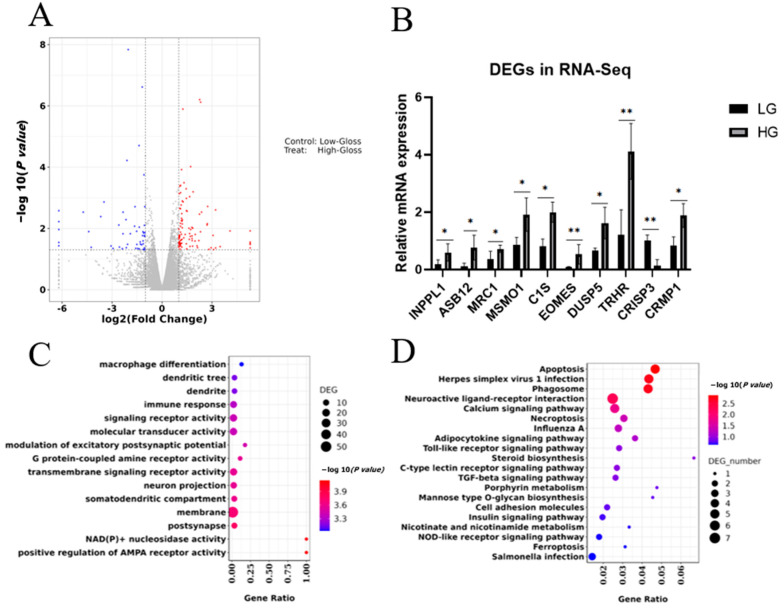
(**A**) The volcano plot maps of all DEGs between HG and LG uteruses. Red dots represent significantly upregulated genes and blue dots represent significantly downregulated genes. (**B**) Ten DEGs were validated by qPCR. Symbols “*” and “**” indicate a significant difference at *p* < 0.05 and *p* < 0.01, respectively. (**C**) GO enrichment analysis of DEGs. (**D**) KEGG pathway enrichment analysis of enriched DEGs. Abbreviations: DEGs, differentially expressed genes; HG, high gloss; LG, low gloss; GO, Gene Ontology.

**Figure 4 animals-14-01141-f004:**
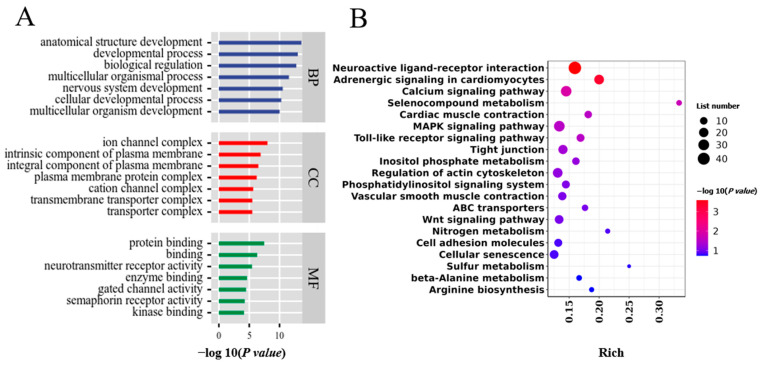
GO enrichment analysis and KEGG pathways of SNP sites located within genes. (**A**) GO enrichment analysis in BP, CC and MF. The blue, orange and green columns represent BP, CC and MF, respectively. (**B**) KEGG pathway enrichment analysis. Abbreviations: MF, molecular function; CC, cellular component; BP, biological process.

**Figure 5 animals-14-01141-f005:**
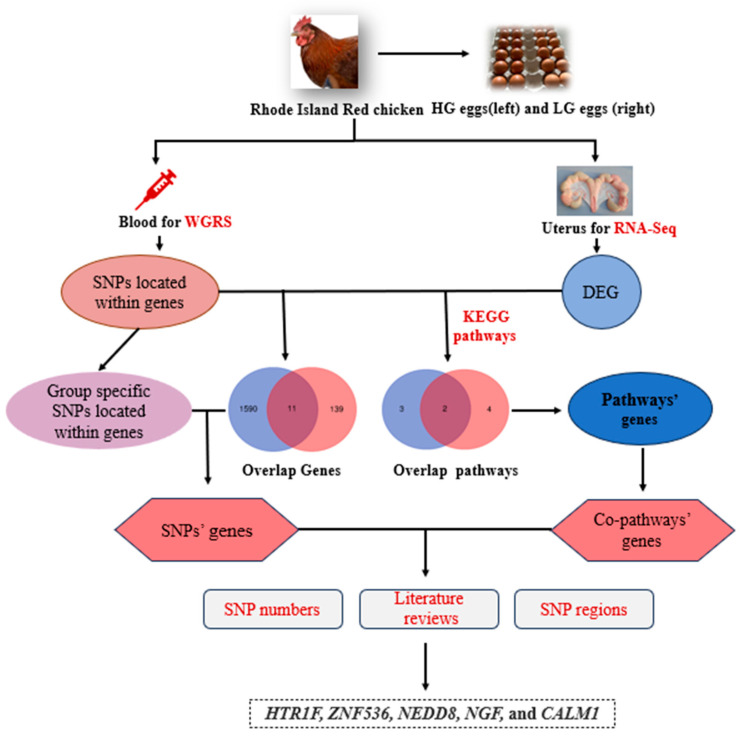
The flow of screening the potential candidate genes affecting eggshell glossiness in Rhode Island Red chicken. Abbreviations: HG, high gloss; LG, low gloss; DEGs, differentially expressed genes; WGRS, whole-genome resequencing.

**Table 1 animals-14-01141-t001:** Top 10 differentially expressed genes between HG and LG groups.

Name	log2FoldChange	*p* Value	Chr	Regulation (HG vs. LG)
FGF19	−2.03066	1.45 × 10^−8^	5	Down
SOUL	−1.18806	2.43 × 10^−7^	5	Down
ENSGALG00000031427	2.258618	6.25 × 10^−7^	3	Up
ENSGALG00000009479	2.318946	7.55 × 10^−7^	2	Up
C1S	1.2519	1.27 × 10^−6^	1	Up
APOD	−1.3807	1.97 × 10^−5^	9	Down
ZNF536	−2.10433	6.03 × 10^−5^	11	Down
TRHR	1.716454	9.61 × 10^−5^	2	Up
AQP1	1.173698	0.00012	2	Up
PTGFRN	−1.0838	0.000178	1	Down

Note: Chr, chromosome; HG, high gloss; LG, low gloss.

**Table 2 animals-14-01141-t002:** Population-specific SNPs detection statistics and annotation results.

Region	Number	Percentage
Intronic	8125	55.37%
Intergenic	5813	39.62%
Exonic	186	1.27%
Splicing	2	0.01%
Downstream	110	0.75%
Upstream	98	0.67%
Upstream/downstream	13	0.09%
UTR5	63	0.43%
UTR3	261	1.78%
UTR5/UTR3	2	0.01%
Total	15,040	100%

Note: Downstream, 1 kb downstream of the transcription termination site; upstream, 1kb upstream of the transcription start site; splicing, splicing junction 2 bp; upstream/downstream, the mutation is located in both the downstream of one gene and the upstream of another gene; UTR5/UTR3, the mutation is located in both the UTR5 of one gene and the UTR3 of another gene.

**Table 3 animals-14-01141-t003:** Overlapping genes between RNA-seq and WGRS analysis.

Gene Name	*p* Value	log2FoldChange	Chr	SNP Num
*ZNF536*	6.03 × 10^−5^	−2.104328726	11	20
*CRMP1*	0.000392	1.115834887	4	1
*LYVE1*	0.005761	1.194353931	5	3
*CNTN1*	0.016508	1.049463725	1	7
*NOX3*	0.014941	−2.343634297	3	3
*ARHGAP15*	0.020556	1.04431234	7	1
*INPPL1*	0.022631	1.6402217	4	1
*HTR1F*	0.031011	Inf	1	2
*MRC1*	0.040211	1.005384815	2	1
*C17orf58*	0.043432	−1.064341843	18	4
*TFEC*	0.048114	2.81415403	1	1

Note: Chr, chromosome; SNP Num, the number of SNPs.

**Table 4 animals-14-01141-t004:** Genes in co-pathways between RNA-seq and WGRS.

Co-Pathways	Calcium Signaling Pathway	Neuroactive Ligand–Receptor Interaction
RNA-seq	*TACR3*, *TRHR*, *CD38*, *CHRNA8*, *FGF19*	*HTR1F*, *TACR3*, *GRM4*, *CHRNA8*
WGRS	*NGF*, *CALM1*, *PLCD1*, *NOS2*, *STIM2*, *ADCY9*, *CACNA1D*, *GRIN2A*, *PPP3R1*, *CHRM2*, *FGF3*, *CACNA1B*, *ERBB3*, *HTR7*, *CHRM3*, *CAMK4*, *ADRA1B*, *NTSR1*, *PLCB1*, *HTR2C*, *FGF18*, *ERBB4*, *ATP2B1*, *SLC8A3*, *FGF10*, *CHRM5*, *GNAL*, *CCKAR*	*HTR1F*, *RLN3*, *GRID1*, *VIPR2*, *DRD2*, *PARD3*, *GLRA2*, *ADRA2A*, *HCRTR2*, *CHRNA5*, *FSHR*, *GRIK4*, *PTH2R*, *HTR7*, *CHRM3*, *NTSR1*, *THRB*, *GABRA4*, *SSTR4*, *HTR2C*, *CCKAR*, *GRIN2A*, *CHRM2*, *MC4R*, *GABRQ*, *SSTR5*, *ADCYAP1R1*, *GRM3*, *ADRA1B*, *GABRB3*, *CALCRL*, *GRID2*, *LPAR4*, *GLP1R*, *CHRM5*, *GLRB*

Note: Co-pathways, intersection of enrichment pathways of two sequencing results; WGRS, whole-genome resequencing.

**Table 5 animals-14-01141-t005:** SNP sites of candidate genes for eggshell gloss.

Gene	Site	Chr	Region	Base in HG	Base in LG
NEDD8	506704	35	exonic	C	T
	506810	35	exonic	C	T
	505577	35	exonic	G	A
NGF	4069359	26	UTR5	A	G
	4070415	26	UTR5	C	T
HTR1F	92728689	1	intergenic	T	C
	92970994	1	intergenic	T	C
ZNF536	8393350	11	intronic	C	G
	8583435	11	intronic	A	G
	8591108	11	intronic	G	A
CALM1	43490289	5	intergenic	G	A

Note: Chr, chromosome; HG, high gloss; LG, low gloss.

## Data Availability

The datasets generated and analyzed during the current study are available in the NCBI repository, https://www.ncbi.nlm.nih.gov/bioproject/PRJNA981231 (accessed on 7 June 2023) and https://www.ncbi.nlm.nih.gov/sra/PRJNA992581 (accessed on 14 July 2023).

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
