# Peer review of "Integration of Whole-Genome Resequencing and Transcriptome Sequencing Reveals Candidate Genes in High Glossiness of Eggshell"

_animals, 2024, doi:10.3390/ani14081141_

Round 1

Reviewer 1 Report

Comments and Suggestions for Authors

The eggshell glossiness directly affects the egg appearance, a factor for attraction of potential buyers and the marketing value of the eggs. The cuticle is the determinant element for eggshell gloss. This study tries to decode the molecular basis for high gloss and low gloss by transcriptome analysis and whole genome sequence. Although the author may carry on to validate their findings in the future, at this stage, their data reveal the driving genes which associate with the eggshell glossiness. The study design is sound, the analysis adequate and data well present, the manuscript is recommended to publish with minor corrections as following:

1.      In section 3.2, I am wondering do you have pedigree information for those 1127 individuals, that you can estimate the heritability and genetic correlation for glossiness and other traits.

2.       Whether hens produced homologous eggs in terms of glossiness, for example, how many hens produce different degrees of eggs, or how many hens produce eggs in the same degree based on your records. You should include this kind of table or figure to describe.

3.       The authors should estimate repeatability based on the between-group variance and within-group variance. The parameter is critical to know how much influence by temporary environment effects.

4.      Line 127, SNP stands for “single nucleotide polymorphism”, why you didn’t call INDEL as well? They also can be functional.

5.      Table 2, can authors align those different SNPs over human genome, and then check how many of them are located on the promoter, and enhancers based on human ENCODE sources.

6.      Table 5, just wondering three exon SNPs of NEDD8 are synonymous or non-synonymous substitutions?

7.      The resolution of figures should be increased, especially Figure 1B, 1C, and Figure 4.

8.      There are two reference insertion errors, line 45 and 285.

Comments on the Quality of English Language

The manuscript is easy to read through.

Reviewer 2 Report

Comments and Suggestions for Authors

Overview

The current manuscript entitled "Integration of Whole Genome Resequencing and Transcriptome Sequencing to Reveal Candidate Genes in High Glossiness of Eggshells" investigates the genomic mechanisms underlying the eggshell glossiness trait using RNA-Seq and WGS. Candidate genes are identified through the joint results of both technologies. The objectives and rationale are clearly stated, and the statistical analyses generally properly performed and reported.

General Comments and Questions

1. The introduction does not clearly explain how eggshell glossiness affects sales. It is not convincingly argued whether consumers' buying intentions are truly associated with this trait. Any supportive statistics to demonstrate this?

2. The introduction did not provide sufficient information about why uterine tissues and blood are the most relevant for studying this trait, nor does it explain why other tissues are considered irrelevant. Could other factors, such as nutrition, digestion, and relevant biosynthesis processes, play a role? 

Given that the molecular mechanism of its formation is unclear, the authors may need to validate their candidate predictions. 

Overall, the association between candidate genes and the trait of interest is less convincing due to the lack of support in tissue selection and candidates validation.

3. It is unclear how enrichment analyses were performed, including the algorithms/tools used and their relevant technical parameters.

4. There is a lack of consistency in correcting for multiple testing among DEGs, SNPs, and enrichment analyses. This issue needs justification or resolution.

5. Considering shell glossiness is likely a complex trait influenced by multiple genes, it would be beneficial to perform differential co-expression transcriptomic network analysis. Identifying differentially co-expressed modules related to the trait, and integrating WGS data to determine if certain SNPs are associated with specific hub genes in the sub-networks, could offer deeper insights into the mechanisms of this trait.

Specific Comments and Questions

1. Table 2 lacks clear definitions for the features, such as how intron, upstream/downstream, and splicing sites are defined, including their lengths. Moreover, the table contains duplicated features. Clarification is needed on the data/tools used for mapping SNPs to these features.

2. The abbreviation "WGS" is confusing - does it refer to "Whole Genome Resequencing" or "Whole Genome Sequencing"?

Reviewer 3 Report

Comments and Suggestions for Authors

The authors tried to investigate the formation mechanism of eggshell gloss and to identify potential genes through whole genome sequencing and RNA sequencing. The study is of major importance but i have some concerns

Abstract

Further joint analysis of WGS and RNA-seq data revealed that HTR1F, ZNF536, NEDD8. The name of genes should be written in full as first time.

NGF and CALM1 were identified as potential candidate genes that may affect eggshell gloss in HG and LG eggs, which provides a reference for the study of eggshell gloss and lays a foundation for improving egg glossiness in layer breeding.. Please rephrase. It is too long.

Introduction

The introduction is too short. Please provide more information about scientific background as importance of whole genome sequencing and RNA sequencing and its role in improvement of economic traits.

Materials and methods

I think the number of investigated animals is low for both whole genome and RNA sequence analysis. Interpret.

Please provide method of DNA extraction in details as birds have nucleated RBCs. You could provide the exception in nucleated blood.

Results

Some figures are of low quality such as figure 4. Please improve quality.

Please provide information of each abbreviation under each table as in case of table 3 to give informative data. 

Discussion

Too short discussion. Please discuss your results by referring to previous studies used the investigated genes as biomarkers in economic traits in poultry

References 

Please make the style of references in accordance of journal guidelines.

Comments on the Quality of English Language

Minor English editing is required 

Round 2

Reviewer 2 Report

Comments and Suggestions for Authors

The manuscript has been sufficiently improved.

However, there seems to be some missing citations, for example, Hisat2, HTSeq, etc. was not properly referenced/cited.

Please double check and make sure all citations are properly used.
